# Mucosal Adhesive Chitosan Nanogel Formulations of Antibiotics and Adjuvants (Terpenoids, Flavonoids, etc.) and Their Potential for the Treatment of Infectious Diseases of the Gastrointestinal Tract

**DOI:** 10.3390/pharmaceutics15092353

**Published:** 2023-09-20

**Authors:** Igor D. Zlotnikov, Natalya G. Belogurova, Irina V. Poddubnaya, Elena V. Kudryashova

**Affiliations:** 1Faculty of Chemistry, Lomonosov Moscow State University, Leninskie Gory, 1/3, 119991 Moscow, Russia; nbelog@mail.ru; 2Research Laboratory of Aquatic Environment Protection and Ichthyopathology, Saratov State University of Genetics, Biotechnology and Engineering Named after N.I. Vavilov, 410005 Saratov, Russia; poddubnayaiv@yandex.ru

**Keywords:** gastrointestinal diseases, *E. coli* screening model, chitosan nanogel, combined formulation adjuvant + antibiotic, sturgeon, in vivo test

## Abstract

Bacterial infections are usually found in the stomach and the first part of the small intestine in association with various pathologies, including ulcers, inflammatory diseases, and sometimes cancer. Treatment options may include combinations of antibiotics with proton pump inhibitors and anti-inflammatory drugs. However, all of them have high systemic exposure and, hence, unfavorable side effects, whereas their exposure in stomach mucus, the predominant location of the bacteria, is limited. Chitosan and nanogels based on chitosan presumably are not absorbed from the gastrointestinal tract and are known to adhere to the mucus. Therefore, they can serve as a basis for the local delivery of antibacterial drugs, increasing their exposure at the predominant location of therapeutic targets, thus improving the risk/benefit ratio. We have used *E. coli* ATCC 25922 (as a screening model of pathogenic bacteria) and *Lactobacilli* (as a model of a normal microbiome) to study the antibacterial activity of antibacterial drugs entrapped in a chitosan nanogel. Classical antibiotics were studied in a monotherapeutic regimen as well as in combination with individual terpenoids and flavonoids as adjuvants. It has been shown that levofloxacin (LF) in combination with zephirol demonstrate synergistic effects against *E. coli* (cell viability decreased by about 50%) and, surprisingly, a much weaker effect against *Lactobacilli*. A number of other combinations of antibiotic + adjuvant were also shown to be effective. Using FTIR and UV spectroscopy, it has been confirmed that chitosan nanogels with the drug are well adsorbed on the mucosal model, providing prolonged release at the target location. Using an ABTS assay, the antioxidant properties of flavonoids and other drugs are shown, which are potentially necessary to minimize the harmful effects of toxins and radicals produced by pathogens. In vivo experiments (on sturgeon fish) showed the effective action of antibacterial formulations developed based on LF in chitosan nanogels for up to 11 days. Thus, chitosan nanogels loaded with a combination of drugs and adjuvants can be considered as a new strategy for the treatment of infectious diseases of the gastrointestinal tract.

## 1. Introduction

Diseases of the gastrointestinal tract are a common medical problem in modern society. There are a number of diseases of the gastrointestinal tract with different mechanisms: food or chemical poisoning [1,2], chronic [3] or situational digestive disorders, old-age-related disorders [4], bacterial infections [5,6,7,8] and rotavirus infections that lead to gastritis and gastroduodenitis, peptic ulcers of the stomach and duodenum, gastroesophageal reflux disease, pancreatitis, and functional diseases of the gastrointestinal tract [9]. These infections lead to inflammation and disruption of the gastrointestinal tract with more or less pronounced symptoms.

Bacterial infections are one of the significant etiological factors in stomach and duodenum diseases. For example, *H. pylori* can cause gastroesophageal reflux disease and chronic atrophic gastritis [10]. The treatment strategy for *Helicobacter pylori*-related diseases includes a combination of two antibiotics with different mechanisms that prevents the development of resistance (https://www.ehmsg.org/publications (accessed on 9 September 2023)). The first-line treatment includes (1) a proton pump inhibitor (omeprazole or analogues), (2) clarithromycin, and (3) amoxicillin or metronidazole (MN)—at least 7–10 days. The second-line treatment includes (1) a proton pump inhibitor (omeprazole or analogues), (2) a bismuth-based drug, (3) MN, and (4) tetracycline. However, these drugs have an adverse effect on the organism due to the high dosage required. In particular, MN is well absorbed (up to 80%) and metabolized in the liver, causing hepatotoxicity, and in the case of gastrointestinal diseases caused by bacterial infections, it would be better if it remained in the intestine longer and acted on the mucous shell.

Some natural extracts contain substances (allylmethoxybenzenes, terpenoids, etc.) [11,12,13,14,15,16,17,18,19,20,21,22,23,24,25,26] with a good potential as enhancers or adjuvants in combination with antibacterial drugs, including LF and MF [15,17,18,27], so that the latter can be used more efficiently and at lower doses. Also, flavonoids, terpenoids, and some polyphenols may have multiple biological effects, including antiviral, antithrombotic, anti-ischemic, anti-inflammatory, antihistamine, antioxidant [28,29], and even stimulation of mucus formation by stomach cells.

Plant extracts exhibiting an antibacterial effect are the main source of the most important biologically active compounds, in particular polyphenols, flavonoids, terpenoids, etc. Flavonoids (for example, baicalein) have a gastroprotective effect. The gastroprotective effect of flavonoids is associated with an increase in endogenous prostaglandins, a decrease in histamine secretion, the absorption of free radicals and oxygen derivatives. Catechins—the main component of green tea—inhibit *H. pylori* urease [30].

Eugenol (EG) and cinnamaldehyde inhibited the growth of *H. pylori* strains tested at a concentration of 2 µg/mL; moreover, in an acidic environment (gastric juice model), their activity further increased, helping to eliminate bacterial infection more efficiently without developing resistance. These data indicate the potential use of adjuvants in combination with the main components (antibiotics LF or MN) for the treatment of gastrointestinal diseases caused by enteropathogenic bacteria. Moreover, there is no doubt that a combination of antibacterials with a properly selected adjuvant is beneficial in many cases.

For further improvement, one needs to deliver both the drug and the adjuvant to the target destination and release them at approximately the same rate. This is important because the two different chemical species would most likely have different pharmacokinetic properties. A smart delivery/release system could compensate this difference, extending and synchronizing their exposure to the target as much as practical. We figured that in the case of gastrointestinal infections, such delivery systems could be based on chitosan [31,32]. Chitosan, a natural polymer containing D-glucosamine and N-Acetyl-D-glucosamine units, is known for its mucoadhesive properties and capability to enhance the effects of drugs and accelerate tissue regeneration [33,34,35,36].

Chitosan and its derivatives enhance the antibacterial effect of various drugs on Gram-negative bacteria and Gram-positive bacteria due to adsorption on the cell surface and increased influx [10,37]. It is known that one of the important mechanisms of bacterial infection is the interaction of bacterial oligosaccharides located on the outer cell wall with glycoproteins and epithelial mucins of the mucous membrane [38,39]. In this regard, an important role in suppressing this activity of adhesives belongs to natural polysaccharides, such as chitosan. Polysaccharides can interact with bacterial patterns and prevent infection of the gastrointestinal mucosa [40].

Inflammatory diseases of the gastrointestinal tract can also be caused by a number of bacteria, such as *Clostridioides difficile* in ulcerative colitis and invasive adhesive *E. coli* in Crohn’s disease, etc. [41]. Therefore, *E. coli* can be used as an appropriate model for primary screening of the effectiveness of the new formulations being developed, before researching the real serious pathogens. *E. coli* is a Gram-negative bacterium, a common inhabitant of the human gastrointestinal tract, and enteropathogenic *E. coli* cause gastrointestinal diseases, inflammation and diarrhea.

Here, we have attempted to develop a mucosal-adhesive drug formulation for the potentially more safe and efficacious treatment of gastrointestinal tract infections: a combination of levofloxacin (LF) [42,43,44,45,46,47,48,49,50,51,52], a powerful fluoroquinolone in the complex, with adjuvants included in chitosan-based systems for mucoadhesiveness and prolongation of delivery. This will presumably suppress absorption and high systemic exposure and, hence, unfavorable side effects, whereas their exposure in stomach mucus, the predominant location of the bacteria, will be reinforced.

Aquacultures, such as sturgeon, are prone to various infections, and there is an unmet need to develop a therapeutic feed formulation that would be efficient at delivering the therapeutic payload. This task is far from trivial—it is not possible to just mix antibacterial to the feed. Sturgeon feeds on bottom sediments, so by the time a normal feed formulation sinks to the bottom, much of the antibacterial drug would be lost. Moreover, antibacterials such as LF/MF have a bitter taste, thereby distracting the fish. Therefore, we have attempted to develop a smart antibacterial drug formulation (with adjuvant and delivery/extended-release system) which is miscible with feed, does not lose the therapeutic payload while in water, and masks the bitter taste so that it is edible for the sturgeon. At the same time, we used sturgeon as an in vivo model to obtain the proof-of-concept data, which can be further applicable for humans.

Thus, this paper presents an experimental basis for creating a combined formulation for the treatment of gastrointestinal diseases based on several components: main drug LF/MN or miramistin (MM) enhanced with salicylhydroxamic acid (SHA, a powerful and irreversible inhibitor of the urease enzyme of various bacteria) and plant extracts (EG, menthol, baicalein, limonene, linalool as efflux inhibitors, membrane-penetrating enhancing agent, antioxidant, tissue-regenerating agent).

## 2. Materials and Methods

### 2.1. Reagents

Chitosan oligosaccharide lactate 5 kDa (Chit5), methyl-β-cyclodextrin (MCD), baicalein, and (+)-limonene were purchased from Sigma Aldrich (St. Louis, MI, USA). Menthol and linalool were purchased from Carl Roth GmbH (Karlsruhe, Germany). Eugenol (EG) and menthol at the highest commercial quality were purchased from Acros Organics (Geel, Belgium). Levofloxacin (LF) was purchased from Zhejiang Kangyu Pharm Co Ltd. (Hangzhou, China). Miramistin (MM) enhanced with salicylhydroxamic acid was kindly provided by the Institute of Organic Chemistry of the Russian Academy of Sciences (Dr. Krylov S.S.). Organic solvents, salts and acids were purchased from Reakhim (Moscow, Russia). Components for LB medium were bactotrypton, agarose and yeast extract (Helicon, Moscow, Russia), and NaCl (Sigma Aldrich, St. Louis, MO, USA).

### 2.2. Preparation of β-Cyclodextrin Inclusion Complexes

The preparation of methyl-β-cyclodextrin (MCD) inclusion complexes was performed as earlier described [18].

### 2.3. Nanoparticles Obtaining and Characterization

Using CD spectroscopy (Jasco J-815 CD Spectrometer, JASCO Corp., Tokyo, Japan), the degree of deacylation of Chit5 was determined by the peak at 215 nm corresponding to the absorption of the amide bond, and it was 92–95%.

Chit5 nanoparticles were obtained after 1 h incubation of 5 mg of Chit5 (PBS, pH 7.4) and drug-MCD inclusion complex (5 mg on drug for 20–35% of mass content in final formulation) followed by extrusion (three times through 200 or 400 nm membrane, Avanti Polar Lipids, Alabaster, AL, USA).

Chit5–genipin (Chit5-gen) nanoparticles were obtained after 24 h incubation of Chit5 nanoparticles (5 mg on Chit5) with 0.1 mg of genipin (dissolved in 10 µL of EtOH) [53,53,54,55,56,57].

Particles’ hydrodynamic diameter sizes and ζ-potentials were measured using a Zetasizer Nano S «Malvern» (Malvern Instruments Ltd., Malvern, UK). The topography, phase and magnitude signal images of the nanogels deposited onto freshly cleaved surface of mica were obtained by atomic force microscopy (AFM) using a scanning probe microscope NTEGRA Prima (NT-MDT Spectrum Instruments, Moscow, Russia).

### 2.4. FTIR Spectroscopy

FTIR spectra of samples were recorded using a Bruker Tensor27 spectrometer (Bruker Optics, Ettlingen, Germany) equipped with a liquid nitrogen-cooled MCT (mercury cadmium telluride) detector, as described earlier [15,37,58,59].

### 2.5. UV Spectroscopy

UV spectra were recorded on the UltraSpec 2100 pro spectrometer (Amersham Biosciences, Uppsala, Sweden) three times in the range of 200–700 nm in a quartz cell Hellma 100–QS with an optical path of 1 cm.

### 2.6. Drug Capacity of Nanogels

The drug capacity of nanogels was determined using UV and FTIR spectroscopy using calibration spectra. The mass content of the drug was varied from 20 to 35%.

### 2.7. Study of the Mucoadhesive Properties of Nanogels

To study the mucoadhesive properties, a film of mucin of 1 mg per cell of a culture tablet was used; then, 200 µL of nanogel (0.5 mg/mL) was added. The UV and FTIR spectra of the solution over the mucin film were recorded at certain intervals, and the amount of adsorbed substance was determined.

### 2.8. Antioxidant Activity Using ABTS Assay

The antioxidant activity of drug formulations was determined by using an ABTS assay as described in papers [29,60,61]: registering an absorbance at 734 nm of ABTS cation-radical mixed with drugs.

### 2.9. Antibacterial Activity Studies: FTIR Spectroscopy, Microbiology

The strains used in this study were *Escherichia coli* (ATCC 25922) from the National Resource Center Russian collection of industrial microorganisms SIC “Kurchatov Institute”) and Lactobacillus (*Lactobacillus plantarum* 8P-A3, *Lactobacillus fermentum* 90TC-4, *Lactobacillus casei*) from the commercially available drug Lactobacterin. The culture was cultivated for 18–20 h at 37 °C to CFU/mL ≈ 4 × 10^7^ (determined by A_600_) in liquid nutrient medium Luria–Bertani (pH 7.2) with stirring at 100 rpm.

The FTIR spectra of cells samples’ suspension were recorded using the following procedure: overnight, cell suspensions were washed twice with sterile PBS (pH 7.4) from the culture medium by centrifuging (Eppendorf centrifuge 5415C, 5 min, 5000× *g*). The cells are precipitated by centrifugation and separated from the supernatant, washed twice and resuspended in PBS (2 × 10^9^ CFU/mL) to register the FTIR spectra. Cell suspensions were incubated with drug samples, and FTIR spectra were registered at 37 °C online.

Microbiologic studies. The culture was cultivated for 18–20 h at 37 °C to CFU ≈ 0.3 × 10^8^ (colony-forming unit) in liquid nutrient medium Luria–Bertani (pH 7.2). The experiments in liquid media were conducted by adding 50 μL of the samples to the 5000 μL of cell culture. The specimens were incubated at 37 °C for 8 days. At the specific time, 300 μL of each sample was taken, diluted with PBS, and the absorbance was measured at 600 nm (with CFU control on Petri dishes). For quantitative analysis of the dependences of CFU (cell viability) on the concentration of MF, 50 μL of each sample was diluted 10^6^–10^8^ times and seeded on the Petri dish. Dishes were placed in the incubator at 37 °C for 24 h. Then, the number of the colonies (CFU) was counted. The number of living cells was additionally determined by the fluorescence of the DAPI dye relative to obviously dead cells and living ones after 10 min of incubation of 200 µL of a cell suspension sample with 1 µg/mL of DAPI.

### 2.10. In Vivo Experiments on Sturgeons: Wound Healing and Microbiotic Composition of the Intestine

According to the principle of analog groups, experimental groups (control, wounded fish with and without treatment) were formed from 40 hybrid individuals of fingerlings of Russian and Siberian sturgeon with an average weight of 110 g, and 10 specimens were placed in four aquariums with a volume of 250 L each. In the formed experimental groups, tagging was performed by trimming the fins. After the formation of experimental groups, an equalizing period of experience lasted for 3 weeks, during which the fish became used to the new conditions of keeping and feeding, and their behavior and feed consumption were studied.

Experimental group received food with chitosan–MCD–nanogels containing dosages of levofloxacin: 4.1 mg per 1 kg of fish weight for treatment for 10–12 days. During the accounting period, indicators of productivity, feed consumption, and the functional state of internal organs were determined, and the biochemical parameters of blood and antimicrobial activity of the drug were studied.

Control weighing of the fish was carried out to determine the dynamics of growth; blood was taken from the heart muscle to analyze biochemical parameters and flushes from wounds. During the same time periods, control of the parameters of experimental individuals of 3 individuals from each group was carried out to study the state of muscle tissue and internal organs, and intestinal contents were selected for sowing on media with subsequent analysis of microflora.

## 3. Results and Discussion

A promising antibacterial drug formulation implies the main drug (LF—levofloxacin, MM—miramistin, or MN—metronidazole) enhanced with adjuvants (allylbenzenes, terpenes, terpenoids, flavonoids, etc.) in the form of inclusion complexes with MCD (for solubilization and controlled release of otherwise insoluble adjuvants) in the composition of chitosan-based nanogels (based on Chit5).

The experimental steps necessary for the development of such a composition are given below: (1) obtaining inclusion complexes of antibiotics and adjuvants with MCD, as well as the inclusion of these complexes in nanogels of chitosan, cross-linked or not with genipin; (2) primary screening of antibacterial activity of formulations against *E. coli* ATCC 25922 (as a screening model of pathogenic bacteria) and *Lactobacilli* (as a model of normal microbiome) to optimize the composition of the drug; (3) studying the effect of the most promising preparations in a long-term experiment; (4) the study of the mechanisms of antibacterial action of nanogels using FTIR spectroscopy, which provides information about the structural components of cells, as well as the study of the physicochemical basis of the nanogels formation; (5) study of antioxidant and mucoadhesive properties of drug-containing chitosan nanogels, which are important for preventing bacterial infection and stimulating early healing; (6) studies of the activity of the effect of drugs on sturgeon fish; (7) summarizing the activity of drug formulations in vitro and in vivo and discussing the potential applicability for the treatment of infectious diseases of the gastrointestinal tract.

### 3.1. Inclusion of Drugs in MCD and Chitosan Nanogels

Some antibacterial drugs—and even more so the adjuvants used in this work—are compounds that are poorly soluble in water; therefore, their practical use requires the obtaining of soluble forms. Cyclodextrins (CDs) are often used as molecular containers for drugs because they have a hydrophobic inner cavity and a hydrophilic outer shell [18,47,62,63,64,65,66,67,68,69,70,71]. However, the inclusion complexes formed are not very strong (*K*_d_ 10^−2^–10^−4^ M). Therefore, in this work, drugs were used not just as an inclusion complex with MCD but were additionally packaged in chitosan-based polymer nanoparticles, which further increased the degree of loading and stability of the formulation and also made it possible to use several drug molecules simultaneously. The formation of inclusion complexes and nanogels was monitored using FTIR spectroscopy, which provides information about the microenvironment of functional groups. Figure 1 and Figure A1 show the FTIR spectra of the main components of the drug formulations as well as their adjuvants. Characteristic peaks of drug molecule bond oscillations (ν(C=C—alone double bond or aromatic system) 1450–1650 cm^−1^ depending on the structure, ν(C=O) 1640–1720 cm^−1^, ν(O–H) 3300–3500 cm^−1^) and MCD or Chit5 bonds oscillations (ν(C-O-C) 1000–1100 cm^−1^) are observed in the FTIR spectra of nanoparticles. When drug is included in the MCD cavity, the hydrophobicity of the molecule increases, which is reflected in the shifts of peaks in the FTIR spectra. For example, when MM is included in the MCD cavity, the band of valence oscillations C=C shifts from 1483 and 1467 cm^−1^ to 1460 cm^−1^. After the formation of chitosan nanoparticles, there is practically no additional shift (regarding the complex with MCD), which indicates the inclusion of drug–MCD complexes in the nanogel. After the cross-linking of chitosan chains with genipin, particles are compacted, which is reflected in a low-frequency shift of the specified band up to 1456 cm^−1^.

### 3.2. Molecular Mechanism of Formation of Chitosan–Genipin-Based Nanogels

To increase the degree of inclusion of drugs and retention of chitosan particles with included antibacterial agents in the delivery system, the cross-linking of Chit5 with genipin was performed. The cross-linking of chitosan with genipin involves two different functional groups in genipin, as described elsewhere [53,53,54,55,56,57]. The degree of cross-linking is chosen optimally so that on the one hand, the drug is included and firmly held, and on the other hand, it does not form insoluble aggregates. The molecular mechanism of formation of chitosan-based nanoparticles can be studied in detail using FTIR spectroscopy (Figure 2a). With increasing temperatures, the mobility of polymer chains increases, and the structure of polymer particles changes, which is reflected in the position of the characteristic peaks (Figure 2b). The peak corresponding to the N-H valence oscillations shifts to the high-frequency region from 3420 to 3580 cm^−1^ with the brightest character in the temperature range of 25–30 °C, which indicates the rupture of hydrogen bonds. The peak corresponding to the C=O valence oscillations gradually shifts to the low-frequency region from 1641 to 1633 cm^−1^, indicating an increase in the degree of hydration of carbonyl groups, which is probably due to their exposure to an external solution while laying hydrophobic sections of chitosan polymer chains inside nanoparticles (into core). The peak corresponding to the C-O-C valence oscillations shifts to the high-frequency region from 1080 to 1100 cm^−1^ with a sharp transition in the temperature range of 40–47 °C following the compaction of particles at elevated temperature. Thus, the general mechanism of the formation of nanoparticles is as follows: the breaking of hydrogen bonds at 25–30 °C, a gradual increase in hydration, and an exposure of hydrophilic areas to the outside, which is followed by compaction of the nanoparticle core. The study of the behavior of chitosan nanoparticles with a drug is important, since their physiological and chemical properties (zeta potential, size, mucoadhesiveness, antibacterial properties) differ significantly from simple polymers.

The formation of a gel nanoparticle is confirmed by flow cytometry data for chitosan particles cross-linked with genipin and loaded with FITC (Figure 2c,d). We have previously proposed the flow cytometry method as a way to characterize nanoparticles and their effects on cells [72]. Side scattering is approximately equal to front scattering (Figure 2c,d) for most particles in the population; this really indicates the formation of large nanoparticles but not debris.

### 3.3. Antibacterial Activity of Drugs in a Free Form and in the Form of a Nanogel

#### 3.3.1. Broad Activity Screening of Individual Components of Drug Formulations

Table 1 presents data on the bacterial growth-inhibiting ability for various forms of drugs, and it also shows the selectivity coefficients of the action against *E. coli* vs. *Lactobacillus* as a model (or surrogate) of the therapeutic window. Among the candidates for the main components of the formulation, LF is the most potent, MM and SHA are of medium potence, and MN is weakly potent. At the same time, it is worth noting the high selectivity coefficients against *E. coli*, especially for LF and MM in the form of nanogels in comparison with other drugs. Of the adjuvants, the most active are zephirol, baicalein, medium-active linalool and EG in the form of nanogels. EG and zephirol are characterized by selectivity coefficients greater than 1, which means their use is justified. Myristicin and limonene show good results both in free form and in nanogel. Thus, the most effective drug formulation has the following variable composition: main component (LF/MN/MM/SHA) + adjuvant (EG/baicalein/zephirol/quercetin) in Chit5–genipin nanogel. Genipin crosslinking with chitosan chains allows obtaining a nanogel from a volumetric gel with the simultaneous compaction of particles and an increase in the effectiveness and selectivity of antibacterial drugs.

#### 3.3.2. Combined Formulations LF + Second Drug in Nanogel

Table 2 shows the coefficients of synergy of antibacterial activity, levofloxacin and the second component in the composition of chitosan nanogels in comparison with single substances. Menthol, quercetin, dihydroquercetin, and SHA in combination with LF enhance each other’s action against *Escherichia coli* by a factor of 1.3–1.7 times. Linalool, zephirol, EG, baicalein, azaron, MN and SHA weaken the effect of LF against “good” *Lactobacilli*. Thus, LF formulations with linalool, zephirol, menthol, EG, baicalein and SHA are highly specific against *E. coli* and do not destroy “good” bacteria.

#### 3.3.3. Prolonged Effect of Selected Medicinal Formulations

Illustrating the extended release effect of the drug formulation, Figure 3 shows the growth curves of *E. coli* and *Lactobacillus* bacteria in a liquid medium. MN has a weak effect against *E. coli* and is practically not adjuvated by additional components. On the contrary, LF in the form of a nanogel is more active against *E. coli* and less active against *Lactobacilli* than simple LF. In addition, LF in combination with zephirol in nanogel (Chit5–genipin) increases the effect against *E. coli* by about 50% and weakens against *Lactobacilli*. Drugs in the form of nanogels are active for more than a week, which is promising for use in the treatment of difficult bacterial infections, including *H. pylori*. A simple MCD is not enough to significantly enhance the effect of the drug, but nanogel has a powerful effect. It is expected that in the acidic environment of the stomach, a fairly rapid destruction of complexes with MCD will occur, but the nanogel will be active for a long time (at least a week) due to adsorption on the mucous membrane. Prolongation of the drug’s action in the form of nanogels based on chitosan cross-linked with biocompatible (and also biologically active agent) genipin is necessary for targeted drug action in the area of infection and constant maintenance of drug concentration to minimize the development of infection and resistance.

#### 3.3.4. FTIR Spectroscopy and Flow Cytometry for Studying the Mechanism of Drug Action on Cells

FTIR spectroscopy is applicable both for the quantitative measurement of the activity of antibacterial formulations (Figure 4a) and for studying the mechanism of action of the drug and polymer on cells (Figure 4b). Figure 3a shows the FTIR spectra of *E. coli* cells after 24 h incubation with drug formulations: the intensity of peaks of amide I and amide II correlate with the number of living cells. The FTIR spectroscopy data on cell viability are in good agreement with the above data of the microbiological experiment. To determine the mechanism of action of nanoparticles on bacterial cells, the incubation of a bacteria suspension with drug–nanogel was used (Figure 4b). An increase in all characteristic peaks is observed: amide I and amide II peaks indicate the interaction of the drug and polymer with cell proteins; the peak at 1000–1100 cm^−1^ demonstrates the adsorption of the polymer on the cell surface; peaks of C–H valence oscillations (2850 and 2920 cm^−1^) indicate the state of lipid bilayer of the cell. An additional control that the polymer is actually adsorbed on the surface of *E. coli* cells was carried out using flow cytometry (Figure 4c–f): histograms show the appearance of a green fraction in the FITC (loaded into Chit5–genipin nanoparticles) fluorescence intensity distribution, which indicates the adsorption of nanoparticles on the surface of bacteria. Thus, it is shown that the drug in the form of a nanogel is effectively adsorbed on the cell wall and then penetrates inside, which affects the enhancement of the antibacterial effect demonstrated above.

### 3.4. Antioxidant Activity of Components of Natural Extracts and Drugs in Chitosan Nanogel

The antioxidant activity of antibacterial formulation brings the benefit of neutralizing excessive free radicals generated by the immune system to fight bacterial infection. There are several methods for studying antioxidant activity: ABTS, DPPH, and beta-carotene assay [29,61,73]. Here, we use ABTS assay because it provides relevant data on antioxidant activity consistent with other methods. Figure 5 shows the curves of the free radical-scavenging activity of adjuvants and main drugs (LF, MM, MN, and SHA). Table 3 shows the values of IC50 free radical scavenging of the studied substances in nanogels. Quercetin, dihydroquercetin, EG, baicalein, and SHA demonstrate powerful antioxidant activity at concentrations of ~0.01 mg/mL: an almost instantaneous elimination of free radicals is observed. LF, MN, MM, myristicin and azaron demonstrate the average antioxidant activity (0.1 mg/mL concentration is required). Linalool, zephyrol, and limonene were weakly active, and menthol is not active at all. In comparison with the literature data [60], we obtained one to two orders of magnitude increased antioxidant capacity of adjuvants due to their use in the form of nanogels. Chitosan particles cross-linked with genipin enhance the antioxidant properties of adjuvants. Thus, the drug formulation for the treatment of gastrointestinal diseases will specifically neutralize the effect of pathogens neutralization and protect the body.

### 3.5. Mucoadhesive Properties of Chitosan Nanoparticles

The mucoadhesive properties of chitosan nanoparticles are important to ensure delivery of the cargo to the point of action. Additionally, this may inhibit the interaction of pathogens with a mucous layer, which potentially minimizes infection and prevents disease. In other words, chitosan acts as an antiadhesive agent, since it adheres to the mucous membranes itself. Mucoadhesiveness was determined by the amount of sorbed LF or baicalein in the composition of nanoparticles on a mucin coating (Figure 6). Figure 6a shows the FTIR spectra of mucin with adsorbed limonene–MCD in a simple form and in the form of a nanogel. By changing the intensity of the amide I peak (1620–1680 cm^−1^), we can judge the degree of binding of mucin to the drug and/or polymer, which we showed earlier on the concanavalin A [15,17,27]. Based on the visual analysis of the amide I peak, it follows that limonene–MCD and Chit5 without drug are poorly adsorbed on mucin (a slight change in intensity; about 10% is observed). On the contrary, limonene in the compositions of nanogels based on Chit5 and Chit5 cross-linked with genipin is well adsorbed on the surface of the mucosal model (significant changes in the intensity of mucin’ amide I peak by two to three times). Nanoparticles with the drug have physicochemical properties different from those of a free polymer. Quantitatively, the adsorption characteristics of nanoparticles on mucin were studied using UV spectroscopy (Figure 6b). The content of LF or baicalein in the solution above the mucin substrate was determined. Both in an acidic medium (pH 2) and in a neutral one (pH 7.4), effective adsorption of the drug in the composition of polymer particles on the mucin surface is observed, while the adsorption of free LF or LF-MCD is approximately five to six times lower. In addition, the prolonged release of LF and chitosan particles from the gel from the mucin surface is achieved. Thus, we have demonstrated the mucoadhesive properties based on chitosan, which has three positive effects: (1) targeted delivery of the drug to the affected areas of the mucous membrane, (2) antiadhesive effect for pathogenic bacteria, and (3) wound healing due to the intrinsic properties of chitosan. So, there are prerequisites to develop predominantly gastro-gut-restricted formulations of antibiotic + adjuvant with preferential exposure in stomach mucus, which is the predominant location of the bacteria.

### 3.6. Studies of the Antibacterial and Wound-Healing Activity on Sturgeon Fish

We have used sturgeon fish as model organisms to study antimicrobial action in vivo. Also, there is an unmet need to develop such therapeutic feed formulation, since it is not possible to just mix antibacterial to the sturgeon feed. Sturgeon feeds on bottom sediments, so by the time a normal feed formulation sinks to the bottom, much of the antibacterial drug would be dissolved in the bulk water. Moreover, antibacterials such as LF/MF have bitter taste, thereby distracting the fish. Therefore, developing a smart antibacterial drug formulation (with adjuvant and a delivery/extended-release system), which is miscible with feed, does not lose the therapeutic payload while in water, and masks the bitter taste so that it is edible for the sturgeon is far from a trivial task. At the same time, we used sturgeon as an in vivo model to obtain the proof-of-concept data, which can be further applicable for humans. Three experimental groups (control, wounded fish with and without treatment) were studied. The experimental group received food with chitosan-β–cyclodextrin complexes in the dosages of levofloxacin: 4.1 mg per 1 kg of fish weight for treatment for 10 days. During the observation period, indicators of productivity, feed consumption, and the functional state of internal organs were determined, and the biochemical parameters of blood and antimicrobial activity of the drug were monitored.

The first criterion was wound healing and antimicrobial action (to test the effectiveness of the studied LF-containing nanogels in terms of wound healing and anti-inflammatory properties). Table 4 presents data on the total microbial number in fish wounds. In control group 2, wound healing practically does not occur, and the number of microorganisms does not decrease; it even increases slightly. At the same time, group 1 receiving therapeutic feeding is characterized by visual wound healing and a decrease in the number of microorganisms by four orders of magnitude relative to the control. In addition, it is important to note the extension of the effectiveness of the antibacterial formulation for 11 days.

To test the effectiveness of the studied drugs of LF-containing nanogels in terms of mucoadhesiveness and the effectiveness of their functioning in the gastrointestinal tract, the 2nd criterion was used: the content of the intestinal microflora of sturgeons when using nanogels with LF (Figure 7). A decrease in the total microbial number in the colon of injured fish with treatment indicates a long and active effect of LF-MCD in Chit5 for at least 10 days. In addition, the selectivity of the formulations was shown: in wounded fish, the number of *Lactobacilli* in the microbiota decreases (wound affects the overall status of the fish body and the number of lactic acid bacteria in fish gut is indicative), while the use of LF compositions preserves *Lactobacilli* (1–1.5 orders of magnitude difference) due to wound healing. These data correlate well with the results obtained in this work in in vitro systems where we observed the selectivity of the action of *E. coli* compared to *Lactobacilli*.

The changes in CFU/g in the control group of the fish (without wounds and without treatment) throughout the experiment were statistically insignificant. The changes in control throughout the experiment were insignificant and unreliable. It can be assumed that such a pronounced antimicrobial and healing effect was achieved not only due to levofloxacin but also due to chitosan as a delivery system, which in addition is known to have its own antimicrobial, antimycotic and immunomodulatory activities. Thus, the effectiveness and prolongation of the action of antibacterial formulations based on LF, cyclodextrin and chitosan has been shown.

## 4. Conclusions

Chitosan–methyl cyclodextrin (MCD) nanogel has proven to be an efficient carrier for both antibiotics and their adjuvants. Safe and natural components of essential oils (such as terpenoids, flavonoids, allylbenzenes, etc.) were used as promising adjuvants, providing the enhancement of antibacterial, anti-inflammatory, antioxidant, regenerating activity. Additionally, adjuvants are able to inhibit efflux in bacteria, which helps overcome bacterial drug resistance. Here, we have developed a promising drug formulation based on the following components: the main drug (antibiotic) and its adjuvant in the complex with cyclodextrin (MCD) (for solubility) and chitosan for the formation of nanogels that demonstrate improved properties compared to a simple polymer. To increase the degree of inclusion of drugs and retention of chitosan particles with included antibacterial agents in the delivery system, the cross-linking of Chit5 with genipin was performed. We selected four candidates for the main component and 10 additional components (adjuvants) and optimized the composition of the drug formulation in terms of the strength and selectivity of the antibacterial action. *E. coli* cells (primary screening model) as a target and *Lactobacilli* (“good” cells) were considered to study the selectivity of the combined formulation against the target bacteria.

We have shown a pronounced efficacy of such formulations against *E. coli* bacterial cell culture (as a model of pathogenic bacteria); at the same time, the EC50 for *Lactobacteria* (as an indicator of the probiotic microbiome) was substantially higher than the antibiotic alone. So, this formulation not only improves delivery and extends release, it also improves the therapeutic window of the antibiotic, which was not anticipated in the beginning. As confirmed by cytometry data, this result may be caused by chitosan’s enhanced affinity to the *E. coli* cell wall. The best results for the combination of antibacterial activity were shown by samples of levofloxacin with the adjuvants baikalein or zeferol in chitosan nanogels with genipin. The genipin cross-linked chitosan nanogel shows the best adsorption on the surface of the mucous membrane, which can potentially improve wound healing and sustainable release of the drugs. Interestingly, we have found that the IR spectra of the bacterial cultures are indicative of the antibacterial action of the tested drug formulations. Not only has this made our drug screening process radically more efficient, but it also shed the light on the specific cell components most affected by the antibiotic. Furthermore, we have tested the drug formulations in vivo using the aquaculture of sturgeon fish with and without the wound of spinal muscle. Levofloxacin in the chitosan–MCD formulation was added to the feed in therapeutic doses. On day 11, the gastrointestinal microbiota composition was shifted in favor of *Lactobacilli*, which is an indicator of the “correct” probiotic microbiome for the treatment group (compared to the untreated group). The total bacterial count in the wound wash differed by four orders of magnitude (much lower for the treatment group), indicating that the orally administered antibiotic composition was edible for the sturgeon, systemically available and efficient in the treatment of the bacterial infection.

## Figures and Tables

**Figure 1 pharmaceutics-15-02353-f001:**
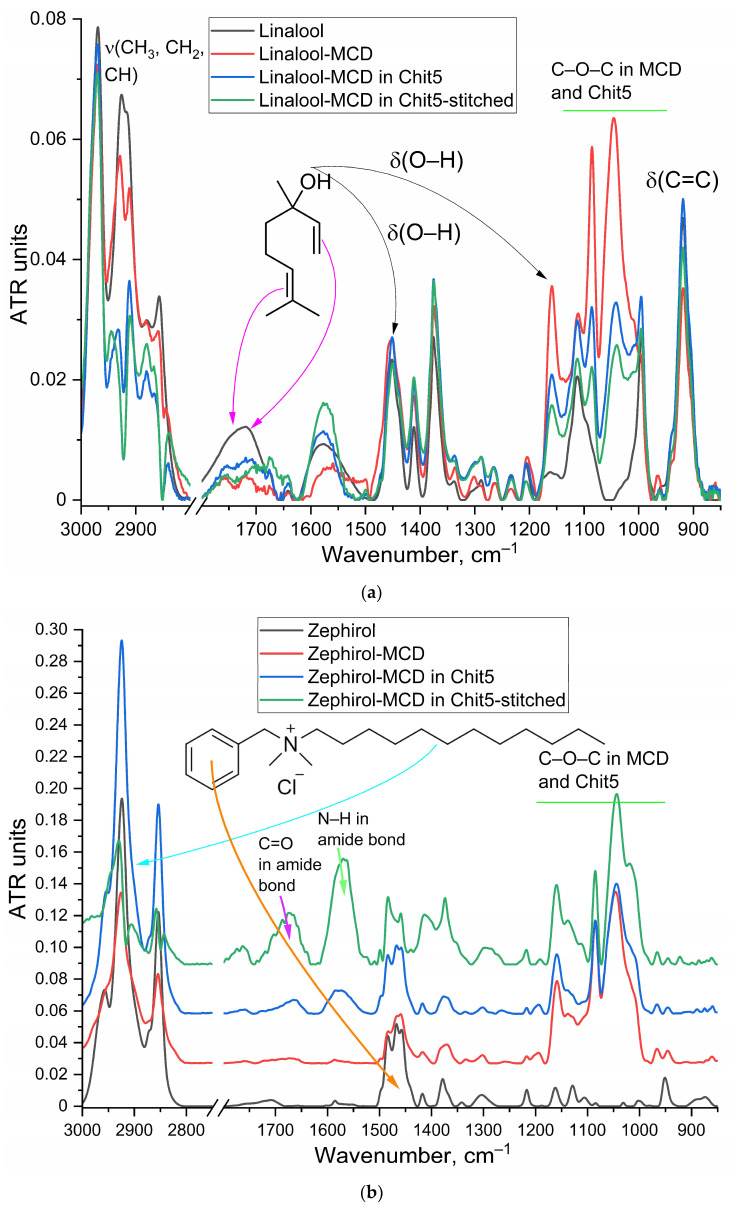
FTIR spectra of drugs in free form, in the form of MCD–inclusion complexes, including wrapped in a Chit5 polymer globule or cross-linked Chit5 with genipin: (**a**) linalool, (**b**) zephirol, (**c**) LF. PBS (0.01 M, pH 7.4). T = 22 °C.

**Figure 2 pharmaceutics-15-02353-f002:**
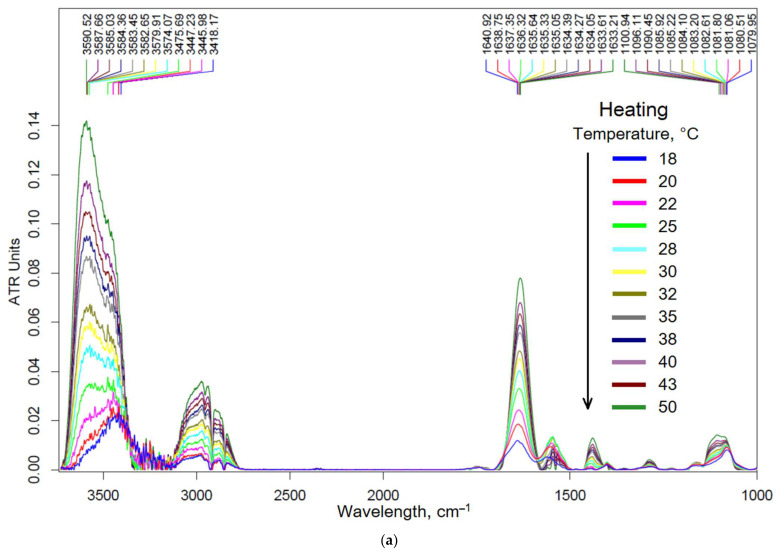
(**a**) FTIR spectra of limonene–MCD in Chit5–genipin particles depending on the temperature–phase transition of chitosan nanogels. (**b**) Corresponding dependences of the position of the characteristic peaks on temperature. (**c**,**d**) Flow cytometry diagrams of Chit5–genipin nanogel with loaded FITC (C_FITC_ = 1 µg/mL). SSC—side scattering, FSC—front scattering, FITC—fluorescence channel.

**Figure 3 pharmaceutics-15-02353-f003:**
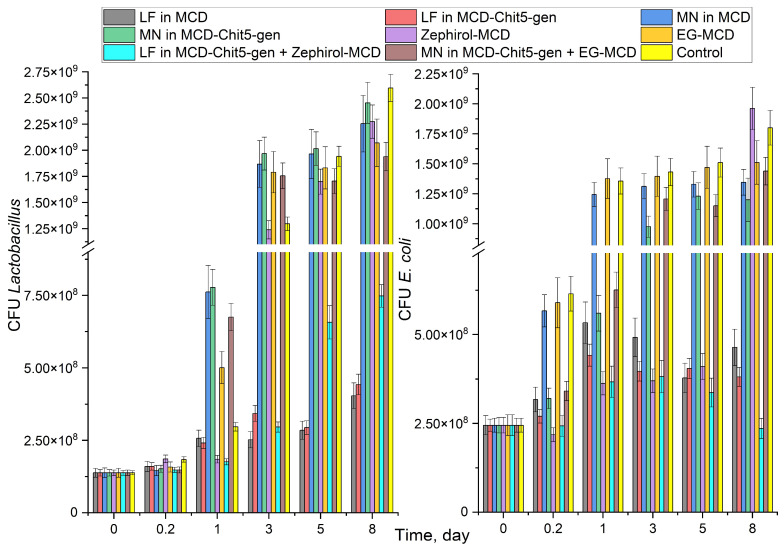
The dependences of *E. coli* and *Lactobacillus* colony-forming units on the incubation time of cells with antibacterial drugs. For *E. coli*: C(LF) = 1 μg/mL, C(MN) = 0.1 mg/mL, C(zephirol) = 0.1 mg/mL, C(EG) = 0.1 mg/mL. For *Lactobacillus*: C(LF) = 10 μg/mL, C(MN) = 1 mg/mL, C(zephirol) = 0.1 mg/mL, C(EG) = 1 mg/mL. LB medium (pH 7.2). 37 °C.

**Figure 4 pharmaceutics-15-02353-f004:**
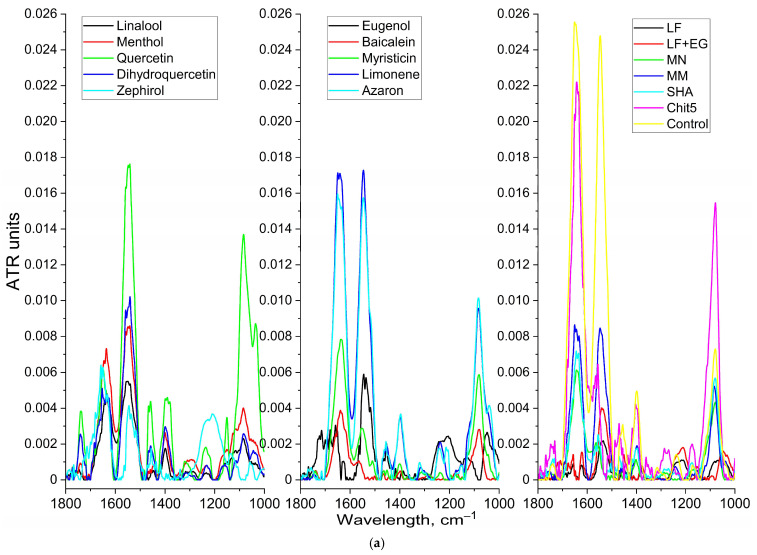
(**a**) FTIR spectra of suspension of *E. coli* cells (10^9^ CFU) after a day of incubation with drug formulations in Chit5 particles. C(LF) = 1 μg/mL, C(MN, MM, SHA) = 0.1 mg/mL, C (other substances) = 1 mg/mL. LB medium (pH 7.2). (**b**) FTIR spectra of suspension of *E. coli* cells (10^9^ CFU) during incubation (online) with linalool–MCD in Chit5 particles. 37 °C. (**c**,**d**) Flow cytometry diagrams of *E. coli* cells incubated with FITC-labeled Chit5–genipin nanogel 15 min (C_FITC_ = 1 µg/mL). (**e**,**f**) Flow cytometry diagrams of *E. coli* cells (control). Green indicates a population with a high intensity of FITC fluorescence. SSC—side scattering, FSC—front scattering, FITC—fluorescence channel.

**Figure 5 pharmaceutics-15-02353-f005:**
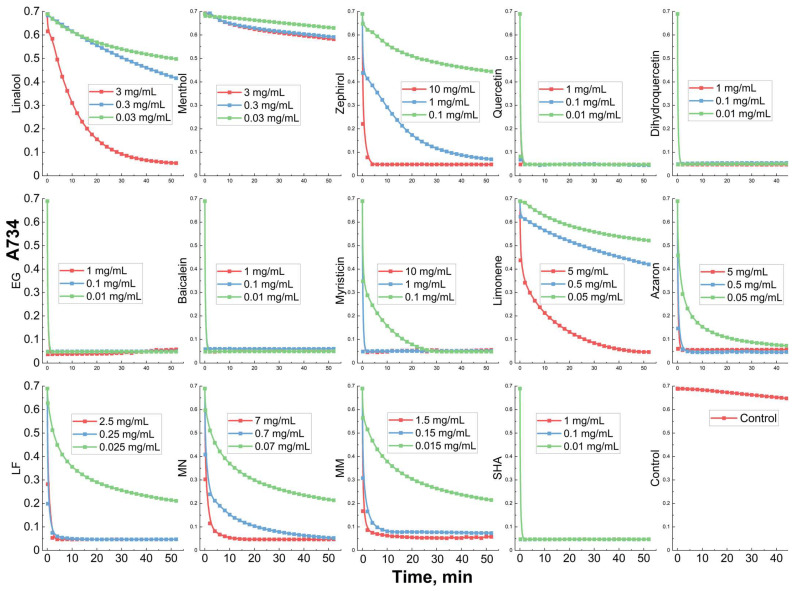
Free radical-scavenging activity of adjuvants and main drugs (LF, MM, MN, SHA) examined by using ABTS assay.

**Figure 6 pharmaceutics-15-02353-f006:**
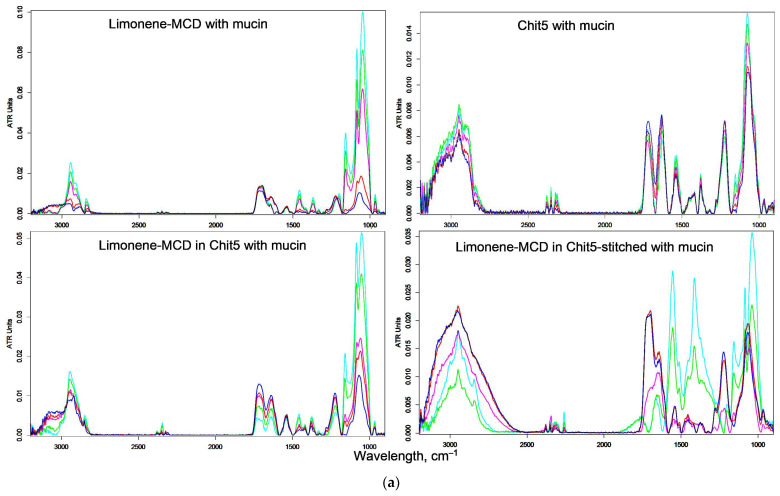
(**a**) FTIR spectra of pre-incubated mucin with limonene-containing formulations and chitosan. pH 2. T = 37 °C. The mass ratio of chitosan/mucin is 0 (blue), 1:30 (red), 1:10 (magenta), 1:3 (green), and 1:1 (cyan). (**b**) Adsorption curves of LF (0.1 mg/mL), baicalein (0.1 mg/mL) and zephirol (0.1 mg/mL) in polymeric particles on a mucin (1 mg) substrate. 0.01 M HCl or 0.01 M Na-phosphate buffer (pH 7.4). (**c**) Release curves of LF and Chit5 from LF-MCD in Chit5–genipin gel on mucin coating. pH 2.

**Figure 7 pharmaceutics-15-02353-f007:**
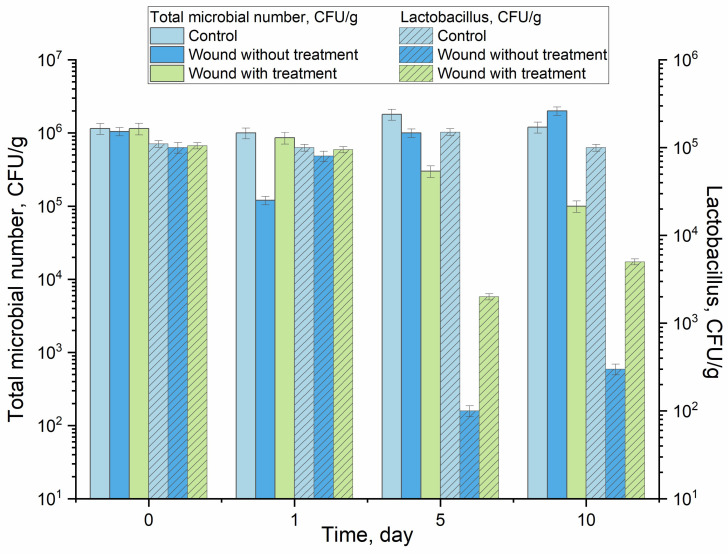
The content of the intestinal microbiota of the colon of sturgeons with wounds in the group with and without treatment as well as the control (without wound).

**Table 1 pharmaceutics-15-02353-t001:** Cell viability (% relative to control) of *E. coli* cells after two days of incubation with samples in LB medium. Selectivity coefficients (the ratio of activities against different strains) of antibacterial formulations (*E. coli* vs. *Lactobacillus*). Underlined numbers indicate the most optimal combinations T = 37 °C.

	in MCD			in MCD-Chit5			in MCD-Chit5-Gen		
	1 mg/mL	0.1 mg/mL	Selectivity (*E. coli* vs. *Lactobacillus)*	1 mg/mL	0.1 mg/mL	Selectivity (*E. coli* vs. *Lactobacillus)*	1 mg/mL	0.1 mg/mL	Selectivity (*E. coli* vs. *Lactobacillus)*
**Linalool**	78 ± 6	85 ± 8	0.6 ± 0.1	73 ± 4	82 ± 5	0.20 ± 0.05	58 ± 3	76 ± 7	0.30 ± 0.06
**Menthol**	83 ± 6	90 ± 4	1.1 ± 0.1	81 ± 5	90 ± 3	0.4 ± 0.1	79 ± 6	84 ± 5	0.4 ± 0.1
**Zephirol**	12 ± 2	34 ± 3	0.8 ± 0.1	10 ± 1	33 ± 2	1.2 ± 0.2	10 ± 1	23 ± 2	1.6 ± 0.2
**Quercetin**	85 ± 5	92 ± 4	1.1 ± 0.1	87 ± 4	92 ± 3	0.9 ± 0.1	82 ± 6	89 ± 5	0.9 ± 0.1
**Dihydroquercetin**	86 ± 3	91 ± 4	1.1 ± 0.1	87 ± 5	95 ± 2	1.0 ± 0.1	83 ± 4	87 ± 6	0.24 ± 0.02
**Eugenol**	79 ± 3	90 ± 4	0.9 ± 0.1	70 ± 2	85 ± 3	1.1 ± 0.1	67 ± 5	82 ± 4	1.2 ± 0.1
**Baicalein**	26 ± 3	72 ± 7	1.8 ± 0.2	49 ± 4	68 ± 6	1.6 ± 0.2	66 ± 3	77 ± 5	0.8 ± 0.1
**Myristicin**	80 ± 5	87 ± 5	1.0 ± 0.1	86 ± 4	86 ± 7	1.2 ± 0.1	90 ± 2	89 ± 3	1.0 ± 0.1
**Limonene**	83 ± 4	90 ± 2	1.0 ± 0.1	87 ± 3	88 ± 5	1.0 ± 0.1	87 ± 6	89 ± 4	0.9 ± 0.1
**Azaron**	81 ± 5	87 ± 4	0.7 ± 0.1	86 ± 6	91 ± 3	0.8 ± 0.1	64 ± 3	87 ± 2	1.0 ± 0.2
	**1 μg/mL**	**0.1 μg/mL**	**Selectivity (*E. coli* vs. *Lactobacillus)***	**1 μg/mL**	**0.1 μg/mL**	**Selectivity (*E. coli* vs. *Lactobacillus)***	**1 μg/mL**	**0.1 μg/mL**	**Selectivity (*E. coli* vs. *Lactobacillus)***
**LF**	9 ± 1	16 ± 3	2.3 ± 0.2	7 ± 1	14 ± 2	3.6 ± 0.3	7 ± 1	13 ± 3	4.0 ± 0.3
	**10 μg/mL**	**1 μg/mL**	**Selectivity (*E. coli* vs. *Lactobacillus)***	**10 μg/mL**	**1 μg/mL**	**Selectivity (*E. coli* vs. *Lactobacillus)***	**10 μg/mL**	**1 μg/mL**	**Selectivity (*E. coli* vs. *Lactobacillus)***
**MN**	41 ± 4	77 ± 5	0.8 ± 0.1	46 ± 7	82 ± 8	0.7 ± 0.1	20 ± 2	69 ± 3	1.4 ± 0.1
**MM**	10 ± 2	26 ± 4	0.7 ± 0.1	9 ± 1	18 ± 3	2.3 ± 0.3	10 ± 1	15 ± 1	2.4 ± 0.1
**SHA**	20 ± 3	84 ± 5	0.5 ± 0.1	20 ± 2	85 ± 3	0.6 ± 0.1	17 ± 2	76 ± 8	0.7 ± 0.1

**Table 2 pharmaceutics-15-02353-t002:** Synergy coefficients (SYC) of antibacterial activity of LF + X in MCD–Chit5 in comparison with alone LF and X in MCD–Chit5 against *E. coli* and *Lactobacillus* cells. The SYC was interpreted as: synergism (>1.2), additivity (0.8–1.2), or antagonism (<0.8). Selectivity coefficients (SEC, the ratio of activities against different strains) of antibacterial formulations (*E. coli* vs. *Lactobacillus*). The SEC was interpreted as highly specific against *E. coli* (>2), highly specific against *Lactobacillus* (<0.5), specific against *E. coli* (>1.3), specific against *Lactobacillus* (<0.85), or indifferent (0.85 < SEC < 1.3). T = 37 °C.

Compound X	Linalool	Menthol	Zephirol	Quercetin	Dihydroquercetin	Eugenol	Baicalein	Myristicin	Limonene	Azaron	MN	MM	SHA
*E. coli*	1.06 ± 0.13	1.49 ± 0.20	1.14 ± 0.05	1.3 ± 0.1	1.3 ± 0.2	1.16 ± 0.07	1.15 ± 0.05	1.08 ± 0.09	1.27 ± 0.08	1.0 ± 0.1	0.23 ± 0.02	1.23 ± 0.05	1.73 ± 0.24
*Lactobacillus*	0.76 ± 0.03	1.11 ± 0.05	0.26 ± 0.02	1.15 ± 0.06	1.06 ± 0.08	0.91 ± 0.05	0.83 ± 0.03	1.24 ± 0.12	1.02 ± 0.06	0.91 ± 0.09	0.69 ± 0.05	1.06 ± 0.07	0.91 ± 0.08
Selectivity *E. coli* vs. *Lactobacillus*	1.4 ± 0.1	1.3 ± 0.1	4.4 ± 0.4	1.1 ± 0.1	1.2 ± 0.1	1.3 ± 0.1	1.4 ± 0.1	0.9 ± 0.1	1.2 ± 0.1	1.1 ± 0.1	0.33 ± 0.04	1.2 ± 0.1	1.9 ± 0.2

**Table 3 pharmaceutics-15-02353-t003:** IC50 values of free radical scavenging of the studied substances in nanogels (Chit5–genipin) examined by using ABTS assay.

Compound	IC50, mg/mL
Linalool	0.46 ± 0.07
Menthol	>3
Zephirol	0.27 ± 0.05
Quercetin	<0.01
Dihydroquercetin	
Eugenol	
Baicalein	
Myristicin	
Limonene	1.5 ± 0.2
Azaron	~0.01
LF	0.015 ± 0.005
MN	0.04 ± 0.01
MM	0.008 ± 0.002
SHA	<0.01

**Table 4 pharmaceutics-15-02353-t004:** Effect of complex formulations based on LF-MCD in Chit5 on the microflora of sturgeon cut wounds.

Group (n = 3)	Total Microbial Number, CFU/g		
	1 day	6 days	11 days
**Group 1.** Fish with wounds and receiving treatment. Basic diet + LF-MCD in Chit5 complex with 23% levofloxacin	(3.0 ± 1.5) × 10^5^	(1.0 ± 0.3) × 10^4^	(1.0 ± 0.4) × 10^2^
**Group 2.** Fish with wounds and no treatment. Basic diet	(4.0 ± 0.4) × 10^5^	(1.7 ± 0.7) × 10^5^	(1.0 ± 0.2) × 10^6^

## Data Availability

The data presented in this study are available in the main text.

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
