# Peer review of "Mucosal Adhesive Chitosan Nanogel Formulations of Antibiotics and Adjuvants (Terpenoids, Flavonoids, etc.) and Their Potential for the Treatment of Infectious Diseases of the Gastrointestinal Tract"

_pharmaceutics, 2023, doi:10.3390/pharmaceutics15092353_

Round 1

Reviewer 1 Report

I have severel methodological concerns that do not allow to recommend publication in current form:

1. E coli was used as a model for H.pylory, which is not correct. Although these are both G- bactreia their biological activities including biofilm formation, sensitivity to antimicrobial drugs etc is very different. Therefore, this may not serve as an appropriate model for H.P. infections and subsequently this is not appropriate work for H.P. infection.

2. E coli is a relevant cause of bacteriemia and sepsis, while enteropathogenic cause diarhea. However, the rationale for developing new treatment for non-existing chronic gut infections by E coli is not correct. Dysmicrobia in IBD is a known factor, however, approaches other than long term admiistration of broad spectrum antibiotics must be used, thus the proposed strategy is only dangerous from the perspective of resistance development.   

3. The aim of the authors was: " a gut-restricted drug formulation for potentially more safe and efficacious treatment of gastrointestinal tract infections". However, there is no single piece of evidence for improved safety or prolonged residence in the gut of the antibiotics in the active form. Moreover, the increased bioavailability/solubility may be expected to worsen the safety compared to standard treatment options. 

4. As there is no evidence for prolonged gut residence of the antibiotics in active form, the presented in vitro data are likely not relevant for in vivo conditions. 

5. Synergy coefficients and antibacterial activity of the adjuvant compounds is not relevant to other than tested E.coli due to different properties of various bacterial pathogens. 

6. there is a clear autocitation missconduct as 12 ot of 64 references used is by the  authors themselves. 

There are also some minor issues to be corrected: 

- Russioan text residuals need to be translated (e.g. line 101)

Reviewer 2 Report

The authors have undertaken a detailed examination of an approach to delivering antibacterial drugs in a complex delivery system for the potential treatment of gastrointestinal infections. The delivery approach involves complexation of drugs with cyclodextrin to improve solubility, combining that material with essential oil components, then trapping the drug complex in a chitosan nanoparticle which may be cross linked with genipin. The purpose of the chitosan is to enable mucosal adhesion to effect localised delivery of the antibiotic.

The work is very detailed, though is primarily focussed on spectroscopy approaches, with incomplete verification of success of the work in accord with the article title. It is suggested the authors  consider the following point in  preparing a revision of their manuscript for further evaluation as to its suitability for publication.

General: the term "stitched" to  indicate "cross-linked" appear several times through the manuscript and should be corrected.

Page 1, lines 1-5: article title is  a little misleading, as gut restirction of the nanoparticle delivery technology is not show in thework. Theauthors have used spectroscopic evidence to  suggest it achieves mucosal adhesion, which might enable gut restriction of the nanoparticles, but good drug absorption of the drug into the systemic circulation might also be attained via solubilisation and  mucosal adhesion.  The title might better substitute "Mucosal adhesive chitosan nanogel..." instead of "Gut-restricted chitosan nanogel...".

Page 3, line 121: section 2.2 title suggests that some discussion of chitosan nanogels production will be described here. This is not done, the section just describes how the degree of deacetylation of the Chit5 was undertaken. Lines 124-126 could be moved to the start of section 2.3, and the inclusion of chitosan nanogels removed from section 2.2.

Page 3, lines 131-132: please give a reference to the method of cross linking Chit5 with genipin, and/or describe how the cross linking  occurs.

Page 5-7, Fig 1:  these spectra might be better included in a supplementary data section  though still referenced and described in section 3.1. That way the figures could be presented   in larger format without having to squeeze them in two columns, with six figures to a page.

Page 16 lines 384- 411: is there  any other validation of the spectroscopic approach to affirming mucoadhesion, other than in the authors' prior work cited here? Is the method shown to predict mucoadhesion compared to physical assessment e.g. of particle retention on mucus against a flowing liquid stream?  Also please further clarify lines 385 - 388, are you saying the chitosan is acting as a mucosal surface adhesive and inhibiting the adhesion of bacteria to mucin?

Page 18, Conclusions:  gastric restriction has not been directly shown so discussion of  how that can be inferred  should be included.  There is no evidence that drug absorption into the bloodstream is controlled by the delivery technology used, so the comments on line446- 448 should be reconsidered or the  conclusion given justified.

Overall the quality of English is good.

Round 2

Reviewer 1 Report

Thank you for good improvement of the manuscript. There are however some minor points left: 

1. The following sence should be still deleted, as it is overoptimistic regading the real clinical situation (line 498-500): "There is a good correlation between the antibacterial activity in vitro and in vivo systems: indicating to the effectiveness and applicability of the developed compositions for the res- toration of intestinal microflora."

2. in the abstract line 35-36  the following claim should be deleted: " and applicable to difficult-to-treat bacteria"  as there is no evidence for difficult  to treat bacteria in the presneted work. 

3. the discussion should include limitation of the work that the claimed decreased systemic absorbtion has not been confirmed (studied) therefore it is a hypothesis. 

4. subsequnetly the aim for claiming this has not been achieved. You should make the aims of the study more precise and further, there is still the clinical claim for improved efficacy and safety, however, this is just expectation, not shown. So this should be deleted from the aims as well. (original point 3). 

Author Response

Dear reviewers and editors! The authors of the presented work sincerely thank you for the study of the article and writing a constructive review! All comments are taken into account. Below is a description of the changes in the work that we made during the audit.

Reviewer's question 1:

Thank you for good improvement of the manuscript. There are however some minor points left:

  1. The following sence should be still deleted, as it is overoptimistic regading the real clinical situation (line 498-500): "There is a good correlation between the antibacterial activity in vitro and in vivo systems: indicating to the effectiveness and applicability of the developed compositions for the res- toration of intestinal microflora."

Answer 1:

We have corrected it

Reviewer's question 2:

  1. in the abstract line 35-36 the following claim should be deleted: " and applicable to difficult-to-treat bacteria" as there is no evidence for difficult to treat bacteria in the presneted work.

Answer 2:

We have deleted it, but we have meant that nanogels-composition developed in the future have the perspectives in the therapy of difficult-to-treat bacterial infection

Reviewer's question 3:

  1. the discussion should include limitation of the work that the claimed decreased systemic absorbtion has not been confirmed (studied) therefore it is a hypothesis.

Answer 3:

We have corrected this point

Reviewer's question 4:

  1. subsequnetly the aim for claiming this has not been achieved. You should make the aims of the study more precise and further, there is still the clinical claim for improved efficacy and safety, however, this is just expectation, not shown. So this should be deleted from the aims as well. (original point 3).

Answer 4:

We have corrected this

Reviewer 2 Report

The authors have considered the feedback from this reviewer and have acted on it where appropriate to revise their manuscript. The submission is very much improved.

Minor literal translations of the authors' mother tongue appear to be present.  These can either be corrected by the editorial team or could be left as-is, because as written the written work still makes sense.

Author Response

Dear reviewers and editors! The authors of the presented work sincerely thank you for the study of the article and writing a constructive review! All comments are taken into account and all necessary corrections have been made.